# Streamlining Language Models via Semantic Basis Analysis

**Yang Li**[*][†]                                                                *yangli1@iastate.edu*
*Iowa State University and Meta*

**Daniel Agyei Asante**[*]                                                       *dasante@iastate.edu*
*Iowa State University*

**Changsheng Zhao**                                                              *cszhao@meta.com*
*Meta*

**Ernie Chang**                                                                  *erniecyc@meta.com*
*Meta*

**Yangyang Shi**                                                                 *yyshi@meta.com*
*Meta*

**Vikas Chandra**                                                                *vchandra@meta.com*
*Meta*

**Reviewed on OpenReview:** *https://openreview.net/forum?id=qq7NNAXvuv*

## Abstract

As the size of language models increases, they deliver substantial performance improvements across a variety of applications. However, this growth also leads to greater computational demands, making deployment on resource-constrained devices—such as personal computers and mobile or wearable devices—more challenging, and significantly raising inference costs on cloud servers. To address these challenges, we introduce Basel, a method to streamline language models by leveraging the semantic structure of their weight matrices. Specifically, Basel treats each weight matrix as a linear combination of bases, selectively retaining those that are associated with essential semantics for the target application, pruning redundant ones, and introducing new bases that enhance task performance. Experimental results demonstrate that Basel achieves significant model size reduction compared to baseline techniques, while maintaining comparable or even superior accuracy across diverse applications.

## 1 Introduction

Large language models (LLMs) (Touvron et al., 2023a; OpenAI et al., 2024; Google, 2023) have significantly enhanced the performance of various applications in natural language processing, computer vision, and beyond. However, their large model sizes pose a bottleneck for many practical uses. The substantial computing resources required for LLM inference make it challenging to deploy them on devices with limited capabilities, such as personal computers and mobile/wearable devices (Li et al., 2024b; 2025). Moreover, even on hardware platforms with ample computing power, deploying LLMs consumes a significant amount of energy, raising concerns about sustainability. Therefore, it is essential to reduce the size of LLMs after pretraining to ease their computational demands and lower energy consumption.

Our approach exploits the relationship between pretrained models and specific target applications. Large language models are typically pretrained on vast datasets encompassing a wide range of tasks, many of which

---

[*]Equal contribution.
[†]Corresponding author. Address: 2434 Osborn Dr, Ames, IA 50011, United States. Email: yangli1@iastate.edu.

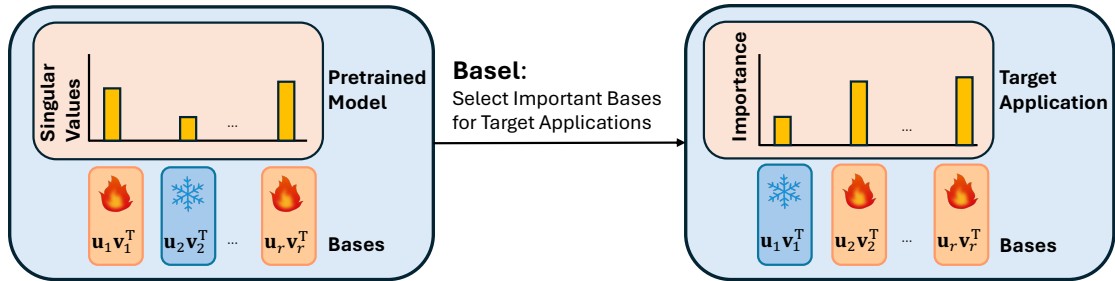

Figure 1: Basel: Identify and select the important bases for target applications during compression.

share common characteristics. This shared pretraining fosters synergies that enhance the performance of LLMs. However, the diversity among these tasks also introduces redundancy into the models. As demonstrated by our results in Section 3, LLMs contain redundant components that are unnecessary for specific target applications. By removing these redundant parts and retaining only the relevant ones, we can reduce the model's size while preserving its performance on the target application. Since many scenarios only require support for a specific type of application (Chang et al., 2024), this approach effectively lowers the computing resource requirements and reduces inference costs.

However, how do we identify the beneficial and redundant components of LLMs for a specific application? In this work, we address this problem through the lens of matrix factorization. Singular value decomposition (SVD) (Golub & Van Loan, 1996) factorizes a weight matrix $\mathbf{W}$ into the product of three matrices $\mathbf{U}$, $\mathbf{S}$, and $\mathbf{V}$, i.e., $\mathbf{W} = \mathbf{USV}^\top = \sum_i s_i \mathbf{u}_i \mathbf{v}_i^\top$, where $s_i$ are (positive) singular values, and $\mathbf{u}_i$ and $\mathbf{v}_i$ are column vectors of $\mathbf{U}$ and $\mathbf{V}$ with unit norms. Our results show that these column vectors $\mathbf{u}_i$ and $\mathbf{v}_i$ may carry specific meanings. For instance, in the LLama 2-7B model (Touvron et al., 2023b), when factorizing the weight matrix $\mathbf{W}_O^h \mathbf{W}_V^h$ of an attention head that is likely useful for code generation tasks, de-embedding the resulting column vectors $\mathbf{u}_i$ and $\mathbf{v}_i$ reveals tokens like `_in`, `<0x0A>`, `_and`, `_to`, and `_for`. These column vectors are evidently useful for code generation but may be less relevant for tasks such as mathematical reasoning.

Inspired by this observation, we propose *Basel*, a low-rank decomposition approach to effectively compress LLMs for target applications. Figure 1 illustrates the key idea of Basel. We view each weight matrix in LLMs as a linear combination of bases $\mathbf{u}_i \mathbf{v}_i^\top$ with singular values $s_i$ as their weights. These bases are valuable representations stored in the pretrained model, learned from large pretraining datasets. For a target application, some bases are advantageous while many others are not. To select the bases beneficial for the target application, we propose retraining the singular values (i.e., the weights of the bases) while keeping the bases fixed, using the training set of the target application. After retraining, we prune the bases associated with small singular values, as they are less important for the target application, and retain only those with large singular values, which are most critical for the target application. This approach allows us to eliminate the redundant parts of the original LLMs and retain only the components essential for the target application. To handle the data distribution differences between the pretraining dataset and the target application, we also augment the model with new bases learned from the training set of the target application during the pruning process. This enables us to learn the new bases necessary for the target application that are absent in the pretrained model.

We evaluate Basel across multiple settings. First, for mathematical reasoning and code generation, we compress Llama 2-7B and Llama 2-13B with Basel and measure pass@1 accuracy on GSM8K (Cobbe et al., 2021) and MATH (Hendrycks et al., 2021) as well as on HumanEval (Chen et al., 2021a) and MBPP (Austin et al., 2021). Compared to the low-rank compression baselines SVD and FWSVD (Hsu et al., 2022), Basel achieves up to 2.7× additional model size reduction while maintaining comparable accuracy on these models and benchmarks. Second, for language modeling, we compress Llama-7B and evaluate perplexity on WikiText-2 (Merity et al., 2016). In this case, Basel yields up to 4× greater size reduction than SVD, FWSVD (Hsu et al., 2022), and SVD-LLM (Wang et al., 2025), while also improving performance. Third, we examine quantization by comparing Basel with the quantization baseline QLoRA (Dettmers et al., 2023),

and we assess pruning by comparing Basel against FLAP (An et al., 2024) and Wanda (Sun et al., 2024) under both quantized and unquantized settings. Finally, we analyze Basel's system-level benefits, showing improvements in token throughput and reductions in memory footprint.

This paper makes the following critical contributions:

- We analyze the relationship between pretrained models and target applications, highlighting the opportunity and underlying rationale for using low-rank decomposition to compress large language models while maintaining performance on target applications.

- We propose Basel, a low-rank decomposition approach to compress pretrained large language models for target applications. Basel identifies the beneficial and redundant components of large language models by relearning the importance (i.e., singular values) of bases using the training set of the target application, and then selects bases based on their importance.

- We evaluate Basel across multiple tasks and models, demonstrating its superior performance in deep compression.

A preprint of this work is available on arXiv (Li et al., 2024a), and the source code of the work is publicly available at `https://github.com/Iowa-State-University-AI-System-Group/Basel` .

## 2  Related Work

Singular value decomposition (SVD) (Golub & Van Loan, 1996) has been applied to reduce the size of machine learning models. Prior research (Xue et al., 2013; Jaderberg et al., 2014; Denton et al., 2014; Zhang et al., 2015; Povey et al., 2018; Chen et al., 2018; Acharya et al., 2019; Noach & Goldberg, 2020) has developed various SVD algorithms to compress different components of models, such as DNNs, CNNs, and embedding layers for a range of applications including natural language processing, speech, and vision. The primary distinction between our work and these prior studies is that they do not *relearn* the importance of bases using the training data of target applications. Instead, they typically prune bases according to the singular values from the original or finetuned models. FWSVD (Hsu et al., 2022) evaluates the importance of individual weights rather than bases during SVD. Considering the importance of bases may improve performance. As shown in Section 4, our approach surpasses FWSVD in deep compression performance.

A recent study (Sharma et al., 2024) applied SVD to large language models. Its focus is on determining the optimal rank for each layer, while our emphasis is on basis selection. The two methods are orthogonal but complementary and can be combined. Chen et al. (2021b); Yu & Wu (2023); Yuan et al. (2023); Wang et al. (2025); Chowdhury et al. (2025) propose reconstructing bases by minimizing the discrepancy between activations before and after compression. These methods are also orthogonal to ours: they concentrate on *basis reconstruction*, whereas we focus on *basis selection*. In principle, their techniques may be integrated with ours to achieve stronger compression performance.

Beyond pruning bases, compression can also be achieved by pruning at other granularities, such as weight pruning (An et al., 2024; Sun et al., 2024) and layer pruning (Hu et al., 2025).

## 3  Basel

In this section, we describe our proposed compression method, Basel.

For a linear layer $\mathbf{y} = \mathbf{W}\mathbf{x} + \mathbf{b}$, Singular Value Decomposition (SVD) factorizes its weight matrix $\mathbf{W} \in \mathbb{R}^{n \times m}$ as the product of three matrices $\mathbf{U}$, $\mathbf{S}$, and $\mathbf{V}$:

$$\mathbf{W} = \mathbf{U}\mathbf{S}\mathbf{V}^{\top} \tag{1}$$

where $\mathbf{U} = [\mathbf{u}_1, \cdots, \mathbf{u}_r]$, $\mathbf{S} = \mathrm{diag}\,(s_1, \cdots, s_r)$, and $\mathbf{V} = [\mathbf{v}_1, \cdots, \mathbf{v}_r]$. The values $\{s_i \in \mathbb{R}, i = 1, \cdots, r\}$ are positive singular values.[1] The vectors $\{\mathbf{u}_i \in \mathbb{R}^n, i = 1, \cdots, r\}$ and $\{\mathbf{v}_i \in \mathbb{R}^m, i = 1, \cdots, r\}$ are orthonormal, i.e., $L^2$ norms $\|\mathbf{u}_i\|_2 = 1$, $\|\mathbf{v}_i\|_2 = 1$, Euclidean inner products $\langle \mathbf{u}_i, \mathbf{u}_j \rangle = \langle \mathbf{v}_i, \mathbf{v}_j \rangle = 0$, for $i \neq j$.

Therefore, we can factorize matrix $W$ as the following series:

$$\mathbf{W} = \sum_{i=1}^{r} s_i \mathbf{u}_i \mathbf{v}_i^\top \tag{2}$$

Let matrix $\mathbf{W}_i = \mathbf{u}_i \mathbf{v}_i^\top$, then

$$\text{Frobenius norm } \|\mathbf{W}_i\|_F = \sqrt{\mathrm{tr}\left(\mathbf{W}_i^\top \mathbf{W}_i\right)} = \sqrt{\mathrm{tr}\left(\mathbf{v}_i \mathbf{u}_i^\top \mathbf{u}_i \mathbf{v}_i^\top\right)} = \sqrt{\mathrm{tr}(\mathbf{v}_i \mathbf{v}_i^\top)} = \sqrt{\mathrm{tr}(\mathbf{v}_i^\top \mathbf{v}_i)} = 1$$

$$\text{Frobenius inner product } \langle \mathbf{W}_i, \mathbf{W}_j \rangle_F = \mathrm{tr}\left(\mathbf{W}_i^\top \mathbf{W}_j\right) = \mathrm{tr}\left(\mathbf{v}_i \mathbf{u}_i^\top \mathbf{u}_j \mathbf{v}_j^\top\right) = 0, \quad \text{if } i \neq j$$

Therefore, $\{\mathbf{u}_i \mathbf{v}_i^\top, i = 1, \cdots, r\}$ can be seen as a group of orthonormal bases in a subspace of $\mathbb{R}^{n \times m}$, and $\{s_i, i = 1, \cdots, r\}$ are their weights, making the weight matrix $\mathbf{W}$ a linear combination of these bases.

This group of bases can be viewed as a series of filters that manipulate the input signal $\mathbf{x}$ to produce the output signal $\mathbf{y}$:

$$\mathbf{y} = \mathbf{W}\mathbf{x} + \mathbf{b} = \sum_{i=1}^{r} s_i \mathbf{u}_i \mathbf{v}_i^\top \mathbf{x} + \mathbf{b} = \sum_{i=1}^{r} s_i \langle \mathbf{x}, \mathbf{v}_i \rangle \mathbf{u}_i + \mathbf{b}$$

In other words, for each basis (i.e., filter) $\mathbf{u}_i \mathbf{v}_i^\top$, the similarity between the input signal $\mathbf{x}$ and the unit direction vector $\mathbf{v}_i$ is measured by their inner product. This inner product is then multiplied by the (positive) singular value $s_i$ to determine the weight for the unit direction vector $\mathbf{u}_i$. The output signal $\mathbf{y}$ is the weighted sum of $\mathbf{u}_i$. Figure 2 illustrates this interpretation of the role of bases from the perspective of signal processing.

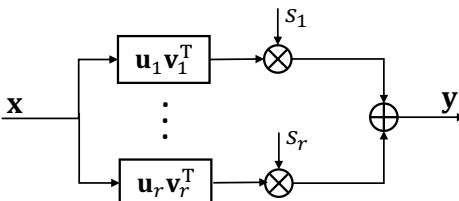

Figure 2: An interpretation of the role of bases from the perspective of signal processing.

Large language models pretrained on diverse datasets learn bases that capture a wide range of semantic concepts. To examine this, we factorize the $\mathbf{W}_O^h \mathbf{W}_V^h$ matrix from the attention layers of both the original Llama 2-7B model and its math-finetuned counterpart. Focusing on bases associated with large singular values, we extract the corresponding $\mathbf{u}$ vectors and interpret their semantics via a de-embedding process. Specifically, we project each $\mathbf{u}$ vector into the vocabulary space by passing it through the output layer (i.e., the lm_head), yielding a logit vector over the vocabulary. We then identify the token with the highest logit as the de-embedding result. Table 1 summarizes our findings. First, some bases are associated with semantic content, such as concepts related to Python, locations, non-English characters, and math symbols. Second, even within the same layer, different bases may correspond to different meanings. For example, in the math-finetuned model at layer 22, attention head 6, basis 1 corresponds to non-English characters, while basis 3 corresponds to math symbols. Revealing the full semantic structure of these bases requires more in-depth analysis, such as the approaches explored in (Oikarinen et al., 2025), which we leave for future work.

---

[1]We drop zero singular values and the corresponding columns of matrices $U$ and $V$.

Table 1: Meaning of bases in the vanilla and math-finetuned Llama 2-7B.

| Domain | Basis | Top ten most probable tokens corresponding to the basis |
|---|---|---|
| Python | Vanilla model, Layer 17, Head 6, Basis 1 | `.` `_in` `<0x0A>` `...` `_...` `_and` `Ľ` `_to` `for` `!` |
| Location | Vanilla model, Layer 17, Head 25, Basis 1 | `_Massachusetts` `_Illinois` `_Chicago` `_Boston` `_Dan` `_Harvard` `_Connecticut` `_IL` `_Bulg` `_Bulgar` |
| Non-English | Math-finetuned model, Layer 22, Head 6, Basis 1 | 學 會 區 國 經 進 : Yā' 無 設 |
| Math | Math-finetuned model, Layer 22, Head 6, Basis 3 | `_{` `{` `}{` `_{r` `={` `_{'` `_{"` `]{` `_'{` `_{};` |

These findings suggest that some bases in the model are useful for some tasks, but may be irrelevant for others. When these irrelevant bases are used as filters in non-target applications, two scenarios can occur: the filter may not be activated (due to a small inner product $\langle \mathbf{x}, \mathbf{v}_i \rangle$), or worse, the filter is activated, introducing harmful information into the output and degrading performance. This indicates that pruning such bases could reduce model size with minimal performance loss, and in some cases, even enhance performance for the target application.

In our approach, Basel, we determine the importance of the bases from the pretrained model by retraining their singular values on the training set of the target application. The weight matrix $\widetilde{W}$ in Basel is represented as:

$$\widetilde{\mathbf{W}} = \sum_{i=1}^{r} \tilde{s}_i \mathbf{u}_i \mathbf{v}_i^\top + \sum_{j=1}^{\tilde{r}} \tilde{\mathbf{u}}_j \tilde{\mathbf{v}}_j^\top \tag{3}$$

In the first term, $\mathbf{u}_i \mathbf{v}_i^\top$ represents the original bases in the pretrained model. To assess their importance, we initialize their weights $\tilde{s}_i$ with their original singular values and then retrain these weights (while keeping the bases fixed) on the training set of the target application. The aim is that, after retraining, the bases important for the target application will have larger singular values, whereas those that are useless or detrimental will have zero or very small singular values. This allows us to identify and prune the less useful bases. Relearning the importance of bases for the target application distinguishes our approach from previous methods. Prior approaches either use the singular values in the original model (Xue et al., 2013; Jaderberg et al., 2014; Denton et al., 2014; Zhang et al., 2015; Povey et al., 2018; Chen et al., 2018; Acharya et al., 2019; Noach & Goldberg, 2020; Sharma et al., 2024) or assess the importance of weight parameters, other than the importance of the bases, to prune them (Hsu et al., 2022). They do not relearn the importance of the bases useful for the target application. From a signal processing perspective, this first term allows us to adjust the weight for each filter, catering to the needs of the target application.

In the second term, the vectors $\tilde{\mathbf{u}}_j$ and $\tilde{\mathbf{v}}_j$ are learnable and serve two main purposes. First, due to distributional differences between the pretraining data and the target application, some bases essential for the latter may be missing. These vectors are used to learn such missing bases. Second, although each pruned basis may have little effect individually, their collective removal can lead to a noticeable performance drop. The additional vectors help mitigate this loss. The number of learnable vectors, denoted $\tilde{r}$, is called the *additional dimension*. From a signal processing perspective, the second term introduces new filters that enhance the model's performance on the target task. While this term adds new parameters, they do not fully contribute to model size growth. This is because SVD is eventually applied to the sum of the first and second terms, reducing the rank if dependencies exist between the newly learned and original, kept bases.

Algorithm 1 presents our proposed approach. Basel operates on a pretrained or fine-tuned model as input. Similar to Hsu et al. (2022); Sharma et al. (2024), Basel iteratively prunes the bases of all linear layers—excluding the embedding layer—including the $\mathbf{W}_Q$, $\mathbf{W}_K$, $\mathbf{W}_V$, and $\mathbf{W}_O$ matrices in the attention layers[2], the linear layers in the feedforward layers, and the output layer. The same compression process is

---

[2]In our implementation, matrices $\mathbf{W}_V$ and $\mathbf{W}_O$ are pruned independently. Exploring alternative strategies, such as jointly pruning $\mathbf{W}_V$ and $\mathbf{W}_O$, is left for future work.

---

**Algorithm 1:** Basel Algorithm

---

**Input:** Pretrained or Finetuning Model $M$
**Output:** Compressed Model $M'$
**Data:** Hyperparameters including KeepRatio, PruningTimes, KeepingEpoch, PruningEpoch,
     PostFineTuningEpoch, $\tilde{r}$

1   IterationsPerPruning = round (NumIterationsPerEpoch * PruningEpoch / PruningTimes);
2   KeepRatioPerPruning = KeepRatio$^{(1/\text{PruningTimes})}$;
3   Convert the weight matrix of each linear layer in $M$—excluding the embedding layer—into the form specified by equation (3);
4   **for** $i = 1$ **to** *KeepingEpoch* **do**
5     |   Tune the learnable parameters in equation (3) including $\tilde{s}_i, \tilde{\mathbf{u}}_j, \tilde{\mathbf{v}}_j, \ i = 1, \cdots, r, \ j = 1, \cdots, \tilde{r}$;
6   **end**
7   **for** $i = 1$ **to** *PruningEpoch* **do**
8     |   Tune the learnable parameters;
9     |   **if** *IterationID is a multiple of IterationsPerPruning* **then**
10     |    |   **for** *each linear layer* **do**
11     |    |    |   Prune bases with smaller singular values $\tilde{s}_i$ such that after pruning, the sum of the singular values of the remaining bases is KeepRatioPerPruning of the sum before pruning;
12     |    |   **end**
13     |   **end**
14   **end**
15   **for** *each layer* **do**
16     |   Compute the low rank matrix $\widetilde{\mathbf{W}}$ based on equation (3);
17     |   $[\mathbf{U}', \mathbf{S}', \mathbf{V}'] = \text{SVD}(\widetilde{\mathbf{W}})$;
18     |   Use two linear layers to substitute for the original layer;
19     |   The first layer's weight matrix is $\mathbf{S}'\mathbf{V}'^{\top}$;
20     |   The second layer's weight matrix is $\mathbf{U}'$;
21   **end**
22   **for** $i = 1$ **to** *PostFineTuningEpoch* **do**
23     |   FineTune the new model $M'$;
24   **end**

---

applied across these components. After each pruning step, the learnable parameters $\tilde{s}_i$, $\tilde{\mathbf{u}}_j$, and $\tilde{\mathbf{v}}_j$ are fine-tuned to compensate for performance degradation. Ultimately, a new weight matrix $\widetilde{\mathbf{W}}$ with a smaller rank $r'$ is learned. We perform a standard SVD on it, representing it as the product of matrices $\mathbf{U}'$, $\mathbf{S}'$, and $\mathbf{V}'^{\top}$. We then replace the original layer with two new layers: $\mathbf{S}'\mathbf{V}'^{\top}$ becomes the weight matrix of the first new layer, and $\mathbf{U}'$ becomes the weight matrix of the second new layer. This reduces the number of parameters from $nm$ to $(n + m)r'$. The new model is subsequently further finetuned to enhance its performance on the target application.

**Overhead Analysis.** Basel updates only the singular values and the newly introduced bases, while keeping the original bases frozen. Unlike full fine-tuning, Basel involves far fewer learnable parameters, substantially reducing training overhead. Table 2 compares Basel and full fine-tuning on Llama-2-7B in terms of GPU hours and memory consumption, evaluated on NVIDIA L40S GPUs (batch size = 32, max sequence length = 512). Under this setting, Basel fits on a single L40S GPU, whereas full fine-tuning requires at least three L40S GPUs due to its higher memory demand. Overall, Basel consumes only about 46% of the GPU hours and 30% of the GPU memory used by full fine-tuning.

Table 2: GPU hours and GPU memory consumption of Basel versus full fine-tuning on Llama 2-7B using NVIDIA L40S GPUs (batch size = 32, max sequence length = 512).

|  | GPU Hours per Batch | Total GPU Memory (GB) |
|---|---|---|
| Basel | $2.87 \times 10^{-3}$ | 40.7 |
| Full fine-tuning | $6.19 \times 10^{-3}$ | 136.2 |

## 4 Experiments

### 4.1 Evaluation Methodology

We evaluate the performance of low-rank compression algorithms on three tasks: mathematical reasoning, code generation, and language modeling. For the mathematical reasoning task, we utilize two evaluation datasets: GSM8K (Cobbe et al., 2021) and Hendrycks' MATH (Hendrycks et al., 2021). The GSM8K dataset comprises verbally described mathematical questions, containing 1,319 samples used for evaluation. The Hendrycks' MATH dataset covers more complex topics such as linear algebra and geometry, consisting of 5,000 question-answer pairs used for evaluation. Due to its complexity, the Hendrycks' MATH dataset necessitates more sophisticated computations and reasoning, resulting in lower accuracy compared to GSM8K. For the code generation task, we use two evaluation datasets: MBPP (Austin et al., 2021) and HumanEval (Chen et al., 2021a). Both datasets evaluate the models' ability to generate Python code. MBPP comprises 500 code generation questions, while HumanEval includes 164 questions. For the language modeling task, we evaluate on WikiText-2 (Merity et al., 2016), a dataset consisting of Wikipedia articles.

We evaluate Basel against low-rank compression methods—SVD, FWSVD, and SVD-LLM—as well as approaches from other compression paradigms. SVD, a widely used technique in prior work on model compression (Sharma et al., 2024; Acharya et al., 2019; Noach & Goldberg, 2020; Xue et al., 2013; Jaderberg et al., 2014; Denton et al., 2014; Zhang et al., 2015; Povey et al., 2018), is applied in our experiments to compress models that have already been fine-tuned on the target task. FWSVD (Hsu et al., 2022) extends SVD by incorporating gradient-based profiling: it estimates weight importance using gradients computed on the fine-tuning dataset and leverages this information during decomposition. SVD-LLM (Wang et al., 2025) reconstructs weight matrices in low-rank form by minimizing the discrepancy between activations before and after decomposition. Beyond low-rank approaches, we also compare Basel with quantization and pruning methods. QLoRA (Dettmers et al., 2023) is a quantization-based technique that improves performance by training LoRA adapters on the fine-tuning dataset. FLAP (An et al., 2024) and Wanda (Sun et al., 2024) are pruning-based methods that prune weights based on weight magnitudes, together with either the magnitude (Wanda) or the variance (FLAP) of input activations profiled on the target task.

Our implementation of Basel is configured with the following key hyperparameters: KeepRatio varies from 70% to 5%, PruningTimes = 100, KeepingEpoch = 1, PruningEpoch = 2 (for math reasoning and code generation) and 1 (for language modeling), PostFineTuningEpoch = 3, and $\tilde{r} = 32$ (see Algorithm 1 for further details).

### 4.2 Evaluating Low-Rank Compression Methods and Pruning Methods for Mathematical Reasoning

Figures 3 (a) and (b) depict the performance of various low-rank compression algorithms on the Llama 2-7B model (Touvron et al., 2023b) for the mathematical reasoning task. We evaluate the models' accuracy

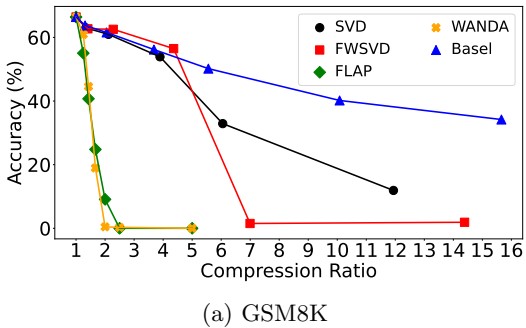
(a) GSM8K

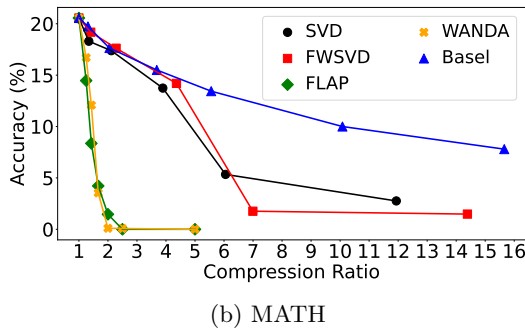
(b) MATH

Figure 3: Pass@1 accuracy and model size of Llama 2-7B compressed with various low-rank algorithms and pruning algorithms on the mathematical reasoning task. Exact values are listed in Tables 4 and 5 in the appendix.

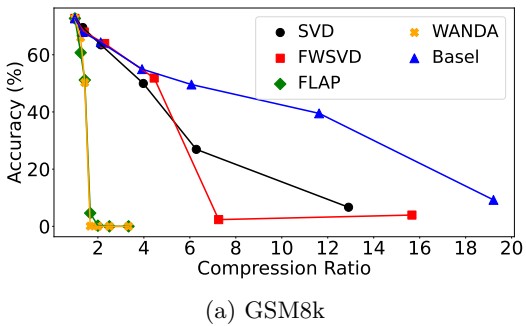 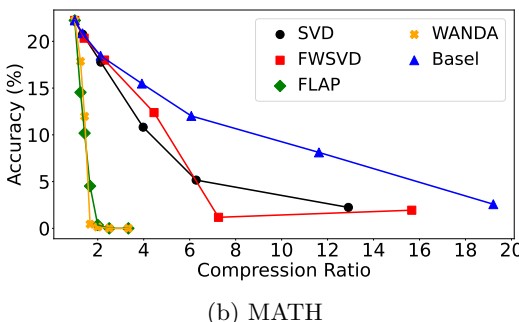

(a) GSM8k  (b) MATH

Figure 4: Pass@1 accuracy and model size of Llama 2-13B compressed with various low-rank algorithms and pruning algorithms on the math reasoning task. Exact values are listed in Table 6 and 7 in the appendix.

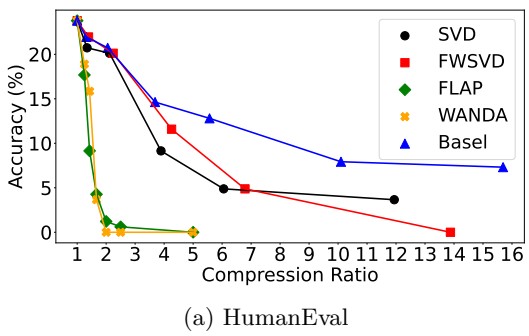 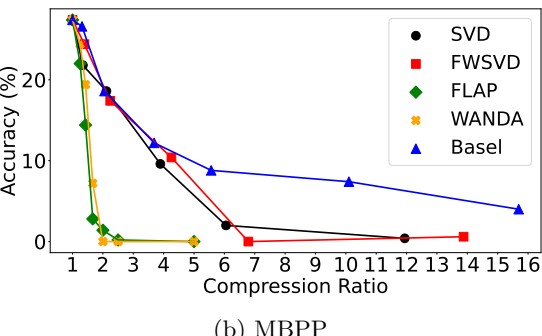

(a) HumanEval  (b) MBPP

Figure 5: Pass@1 accuracy and model size of Llama 2-7B compressed with various low-rank algorithms and pruning algorithms on the code generation task. Exact values are listed in Tables 8 and 9 in the appendix.

(Pass@1) at different compression ratios (original model size vs. compressed model size). For low compression ratios (below 6), all low-rank compression methods achieve similar accuracy. However, our Basel method significantly outperforms SVD and FWSVD at higher compression ratios. For instance, at a 7x compression ratio, Basel achieves around 46% and 12% accuracy on GSM8K and MATH datasets, respectively, while FWSVD drops below 2% accuracy on both datasets and SVD reaches only 27% and 5% accuracy on GSM8K and MATH, respectively. We also find that Basel, at a compression ratio of 16, achieves better accuracy on both GSM8K and MATH compared to FWSVD and SVD at a compression ratio of 6. This suggests that Basel reduces the model size by up to 2.7 times more than SVD and FWSVD while maintaining similar accuracy. This highlights the effectiveness of Basel for deep compression, especially when aiming for aggressive model size reduction.

Similar trends emerge for the larger Llama 2-13B model in Figures 4 (a) and (b). Once again, Basel significantly outperforms SVD and FWSVD at compression ratios exceeding 6. At a 7x compression ratio, Basel achieves 47% and 11% accuracy on GSM8K and MATH datasets, respectively, demonstrating its advantage. This is in stark contrast to SVD's performance (23% and 5% accuracy on GSM8K and MATH) and FWSVD's performance (around 2% on both datasets). These results solidify Basel's effectiveness for deep compression across different model sizes.

Figures 3 and 4 also compare Basel with the pruning algorithms FLAP and Wanda on GSM8K and MATH. The results show that Basel achieves substantially greater model size reduction while preserving comparable or superior performance.

### 4.3 Evaluating Low-Rank Compression Methods and Pruning Methods for Code Generation

Similar results extend to code generation tasks (Figures 5 and 6). For both Llama 2-7B and Llama 2-13B models, all low-rank compression methods perform comparably at lower compression ratios (below 4). How-

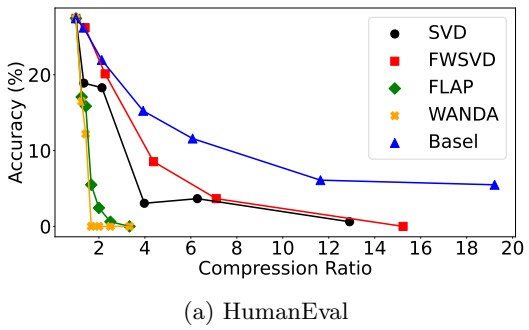

(a) HumanEval

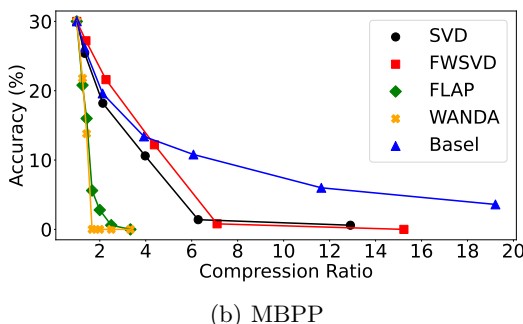

(b) MBPP

Figure 6: Pass@1 accuracy and model size of Llama 2-13B compressed with various low-rank algorithms and pruning algorithms on the code generation task. Exact values are listed in Tables 10 and 11 in the appendix.

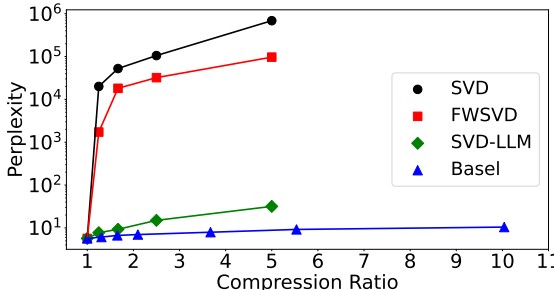

Figure 7: Perplexity and model size of Llama-7B compressed with various low-rank algorithms on WikiText-2. Lower perplexity indicates better language modeling performance. Results for SVD, FWSVD, and SVD-LLM are taken from (Wang et al., 2025). Exact values are listed in Table 12 in the appendix.

ever, Basel exhibits clear superiority at higher compression ratios. On Llama 2-7B at a 6x compression ratio, Basel achieves 12% and 9% accuracy on HumanEval and MBPP datasets, respectively, significantly outperforming SVD (5% and 2%) and FWSVD (6% and 2%). Similar trends hold for Llama 2-13B. Moreover, Basel consistently outperforms pruning baselines FLAP and Wanda, particularly under aggressive compression. These findings further solidify Basel's effectiveness for deep compression across diverse tasks and model sizes.

## 4.4 Evaluating Low-Rank Compression Methods for Language Modeling

For the language modeling task, we compress Llama-7B using Basel and baseline methods SVD, FWSVD, and SVD-LLM, and evaluate the resulting models in terms of perplexity. Perplexity serves as a measure of language modeling quality, with lower values indicating better performance. As shown in Figure 7, Basel consistently yields lower perplexity than SVD, FWSVD, and SVD-LLM at the same compression ratio. Conversely, for a given perplexity, Basel produces a substantially smaller model. For instance, Basel achieves a perplexity of 10.45 at a $10\times$ compression ratio, whereas SVD-LLM reaches a perplexity of 15.00 at only $2.5\times$. This demonstrates that Basel attains up to four times greater compression than SVD-LLM (and even more than SVD and FWSVD) while delivering superior performance.

## 4.5 Combining Low-Rank Compression and Quantization for Mathematical Reasoning

Quantization reduces the precision of weight parameters to shrink model size and is a widely used technique for model compression. When using 8-bit quantization or higher, the performance degradation is typically minimal. However, more aggressive quantization (e.g., 4-bit) often leads to a significant drop in accuracy. We apply both 8-bit and 4-bit quantization to Llama 2-7B and Llama 2-13B models on mathematical reasoning.

For Llama 2-7B, the 8-bit quantized model achieves Pass@1 accuracy of 66.0% on GSM8K and 20.3% on MATH, closely matching the unquantized model's performance (66.4% and 20.6%, respectively, with bf16).

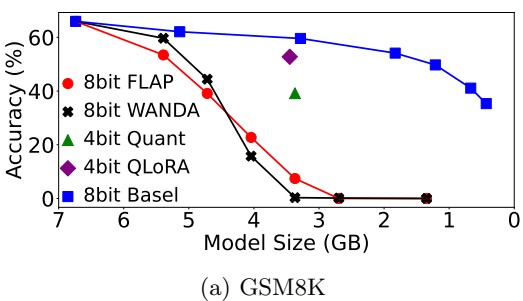 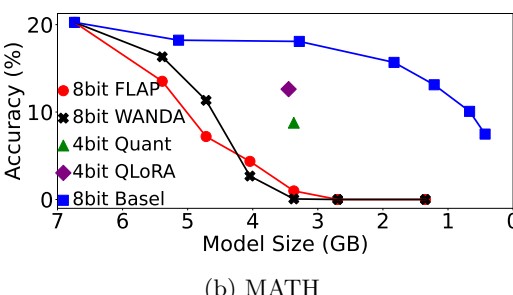

(a) GSM8K · (b) MATH

Figure 8: Pass@1 accuracy and model size of Llama 2-7B compressed using quantization alone, as well as quantization combined with low-rank methods or pruning, on the mathematical reasoning task. Exact values are provided in Table 13 in the appendix.

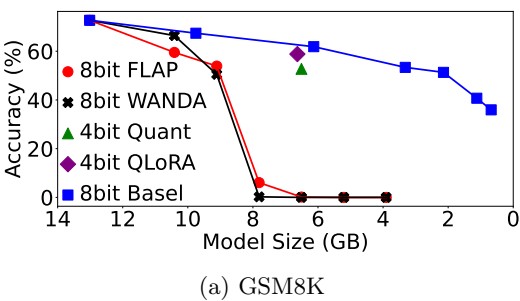 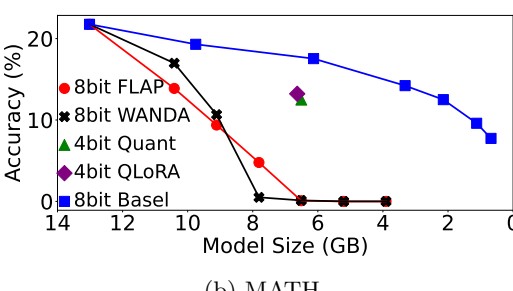

(a) GSM8K · (b) MATH

Figure 9: Pass@1 accuracy and model size of Llama 2-13B compressed using quantization alone, as well as quantization combined with low-rank methods or pruning, on the mathematical reasoning task. Exact values are provided in Table 14 in the appendix.

A similar trend holds for Llama 2-13B, where 8-bit quantization results in only a negligible accuracy drop. In contrast, 4-bit quantization causes substantial degradation. On Llama 2-7B, accuracy drops from 66.0% to 39.2% on GSM8K and from 20.3% to 8.8% on MATH. Llama 2-13B exhibits similarly significant losses. These results suggest that relying solely on quantization—especially at lower bit-widths—can be detrimental to performance.

To address this, we explore combining quantization with low-rank compression using our Basel method. Figures 8 and 9 illustrate that combining Basel with 8-bit quantization not only achieves a smaller model size but also significantly outperforms 4-bit quantization in accuracy. On compressing Llama 2-7B, Basel + 8-bit achieves 59.6% on GSM8K and 18.1% on MATH with a 3.28 GB model size, compared to 4-bit quantization's 39.2% and 8.8% with a larger 3.37 GB model. On compressing Llama 2-13B, Basel + 8-bit yields 61.9% on GSM8K and 17.5% on MATH with a 6.13 GB size, while 4-bit quantization achieves only 52.8% and 12.5% with 6.51 GB.

We further compare Basel + 8-bit quantization against three additional approaches: (1) QLoRA with 4-bit quantization, (2) combining 8-bit quantization with FLAP, and (3) combining 8-bit quantization with Wanda. As shown in Figures 8 and 9, Basel + 8-bit consistently outperforms both in terms of accuracy and size. For instance, on compressing Llama 2-7B, it improves accuracy by 5.5% on both GSM8K and MATH compared to QLoRA, while reducing the model size by an additional 0.13 GB.

### 4.6 Combining Low-Rank Compression and Quantization for Code Generation

We also compare Basel + 8-bit quantization to three alternative methods on the code generation task: (1) standard 4-bit quantization, (2) QLoRA with 4-bit quantization, (3) FLAP combined with 8-bit quantization, and (4) Wanda combined with 8-bit quantization. Figures 10 and 11 present the results for compressing Llama 2-7B and Llama 2-13B. On both models, Basel + 8-bit quantization consistently outperforms 4-bit quantization while achieving smaller model sizes. For example, on Llama 2-7B, it improves accuracy by

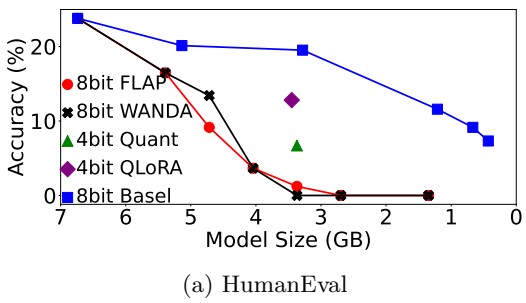

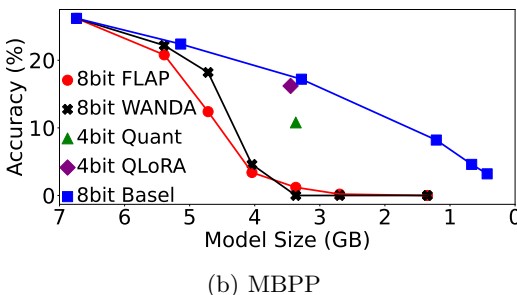

(a) HumanEval

(b) MBPP

Figure 10: Pass@1 accuracy and model size of Llama 2-7B compressed using quantization alone, as well as quantization combined with low-rank methods or pruning, on the code generation task. Exact values are provided in Table 15 in the appendix.

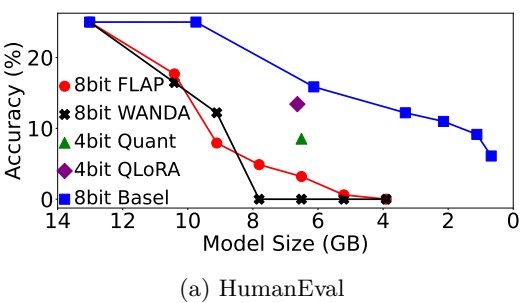

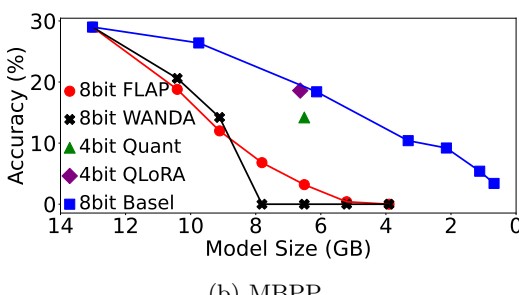

(a) HumanEval

(b) MBPP

Figure 11: Pass@1 accuracy and model size of Llama 2-13B compressed using quantization alone, as well as quantization combined with low-rank methods or pruning, on the code generation task. Exact values are provided in Table 16 in the appendix.

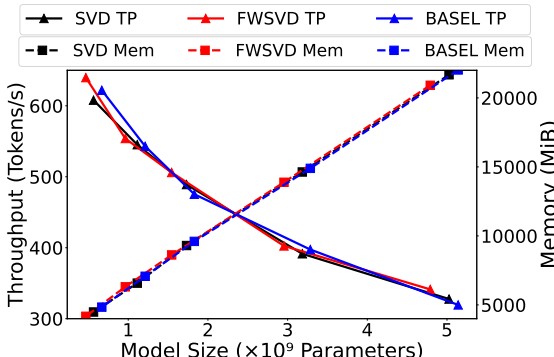

Figure 12: Throughput and memory consumption of compressed models.

12.8% on HumanEval and 6.4% on MBPP, with a model size similar to that of the 4-bit version. This further highlights the benefit of combining low-rank compression with quantization.

Basel + 8-bit quantization also significantly outperforms both QLoRA (4-bit), FLAP + 8-bit quantization, and Wanda + 8-bit quantization on Llama 2-7B and Llama 2-13B. For instance, on compressing Llama 2-7B, it achieves 6.7% higher accuracy on HumanEval and 1.0% higher on MBPP compared to QLoRA, while also reducing the model size by an additional 0.17 GB.

## 4.7 Inference

Figure 12 presents the inference throughput and memory consumption of models compressed from Llama 2-7B on a single A100 GPU, using GSM8K as the evaluation set. The results show that low-rank com-

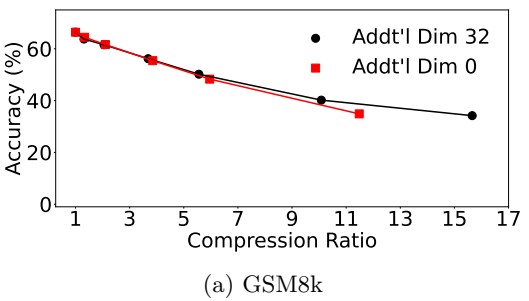 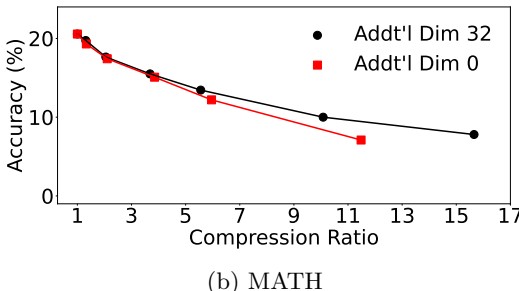

(a) GSM8k  (b) MATH

Figure 13: Ablation study: Impact of varying the additional dimension in Basel on the compression of Llama 2-7B for the mathematical reasoning task. Exact values are listed in Table 19 in the appendix.

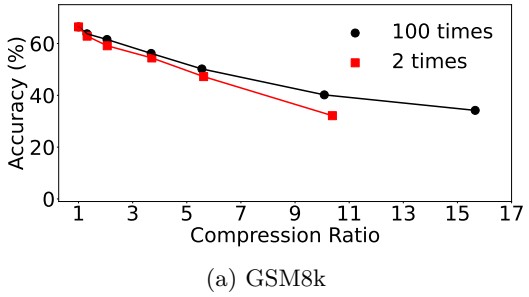 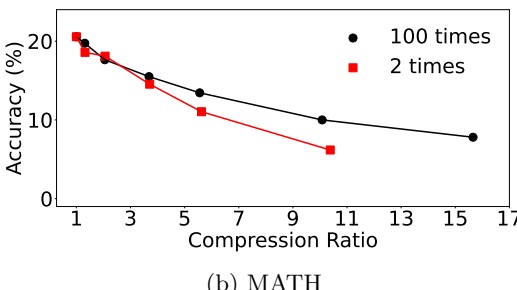

(a) GSM8k  (b) MATH

Figure 14: Ablation study: Impact of varying the pruning times in Basel on the compression of Llama 2-7B for the mathematical reasoning task. Exact values are listed in Table 20 in the appendix.

pression methods, including SVD, FWSVD, and Basel, lead to reduced memory consumption and improved throughput as the model size decreases. Throughput and memory usage are primarily dependent on model size, with no significant differences between the methods at equivalent sizes. However, since our proposed Basel method achieves a greater reduction in model size while maintaining similar accuracy to SVD and FWSVD, it improves throughput by up to 19% and reduces memory consumption by up to 37%.

## 4.8 Ablation Study

We conduct a series of ablation studies to better understand the design of Basel.

**Impact of hyperparameters.** We first evaluate the effects of two key hyperparameters: the additional dimension ($\tilde{r}$ in Equation (3)) and the number of pruning iterations. The additional dimension is introduced to help recover information lost during pruning, particularly under high compression. As shown in Figure 13, setting $\tilde{r} = 32$ improves accuracy when compressing Llama 2-7B for mathematical reasoning, especially beyond a 7× compression ratio. However, this accounts for only a small fraction of Basel's overall performance gain relative to baselines (Figures 3 and 13). Similarly, applying pruning gradually over many iterations allows the model to better adapt to parameter reduction, which is especially beneficial under extreme compression. Figure 14 illustrates this effect: pruning 100 times consistently yields higher accuracy than pruning only twice when the compression ratio exceeds 4.

**Freezing basis vectors.** Basel freezes the basis vectors and updates only the singular values (and any additional bases) during compression. To evaluate this design choice, we compare Basel with a variant that also updates the basis vectors. As shown in Figure 15, freezing the bases results in better accuracy on Llama 2-7B for mathematical reasoning. This strategy also reduces compression time by 33%, since backpropagation involves fewer trainable parameters.

**Inclusion of $L1$ regularization.** Finally, we examine the effect of adding an $L1$ penalty on the learnable singular values $\tilde{s}_i$. The goal is to encourage sparsity in $\tilde{s}_i$, thereby enabling additional compression. We experiment with two regularization weights, $\lambda = 0.02$ and $\lambda = 0.1$. Figure 16 reports the results. With

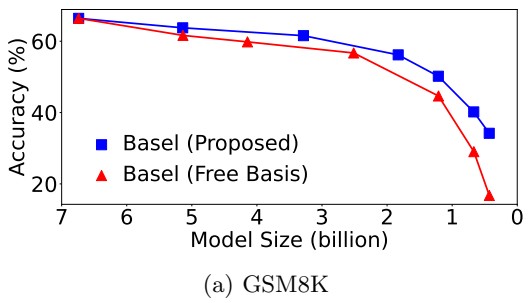 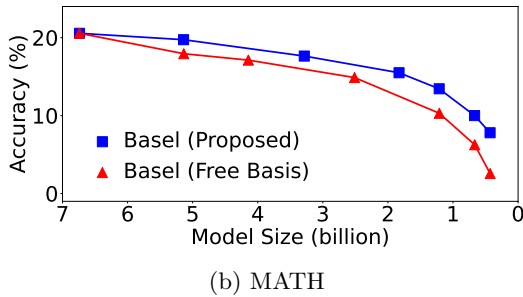

(a) GSM8K

(b) MATH

Figure 15: Ablation study: Pass@1 accuracy and size of Llama 2-7B compressed by Basel (proposed) and Basel (free basis) on mathematical reasoning. Exact values are listed in Table 21 in the appendix.

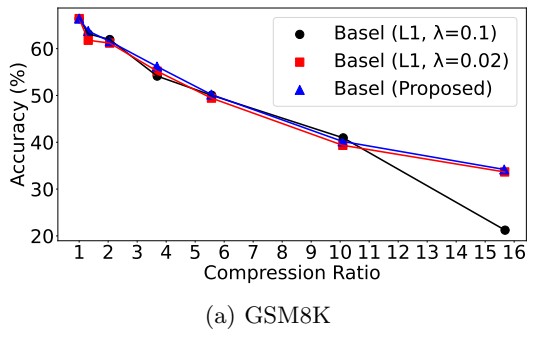 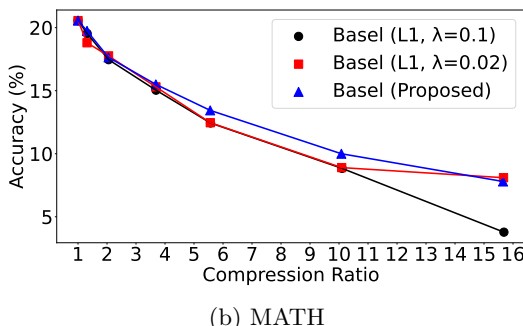

(a) GSM8K

(b) MATH

Figure 16: Ablation study: Pass@1 accuracy and model size of Llama-2-7B compressed with Basel, with and without $L1$ regularization, on the mathematical reasoning task. Exact values are reported in Table 22 in the appendix.

$\lambda = 0.02$, performance is similar to standard Basel; however, a larger weight ($\lambda = 0.1$) degrades performance under deep compression.

## 4.9 Compression vs. Training from Scratch

In many real-world LLM deployment scenarios, compute and hardware constraints impose a target model size tailored to a specific application, which may not match any existing pretrained models. For example, Llama-2 offers 7B, 13B, and 70B variants, but no 4B option. To obtain a model of the required size for the target application, one can either (1) train a new model from scratch or (2) compress a nearby larger model. The first approach is often impractical due to high computational costs—training Llama reportedly requires up to one million A100 GPU hours Touvron et al. (2023a)—and limited access to pretraining data. In contrast, model compression is far more efficient; low-rank methods such as Basel typically require less than 0.01% of the pretraining compute.

To assess the effectiveness of compression, we compare a fine-tuned Llama 3.2-1B model with a compressed version of Llama 3.2-3B reduced to 1B parameters using Basel. As shown in Table 3, the compressed model achieves similar performance on mathematical reasoning. This indicates that, in this setting, compression can

Table 3: Pass@1 accuracy and model size of Llama 3.2-3B compression via Basel for mathematical reasoning.

|                       | 3B   | 1B   | Basel-compressed-3B |
|-----------------------|------|------|---------------------|
| Model Size (billion)  | 3.21 | 1.24 | 1.24                |
| GSM8K Acc (%)         | 72.5 | 54.4 | 55.3                |
| MATH Acc (%)          | 26.1 | 17.6 | 16.7                |

serve as a viable alternative to training from scratch—offering substantial savings in compute and eliminating the need for pretraining data. These results demonstrate the potential of Basel to support efficient model scaling under deployment constraints.

## 5 Conclusion

The significant size of large language models leads to high inference costs and demands substantial computing resources. To mitigate these issues, we focus on compressing large language models to meet the specific requirements of target applications. Our approach involves examining these models through the lens of matrix factorization. By viewing the weight matrix of large language models as a linear combination of a group of bases, we have identified that pretrained models often contain many redundant bases that are less useful for target applications. To address this, we propose Basel, a compression algorithm that evaluates the importance of each base for target applications and prunes those that are less significant. Experimental results demonstrate that Basel significantly outperforms baseline low-rank compression algorithms in achieving deep compression. Basel greatly reduces the inference cost of large language models, making them more accessible and practical for a wider range of applications. This advancement has the potential to democratize the use of large language models, facilitating their adoption and integration across diverse fields and industries.

## Acknowledgments

This research is partially supported by Meta and by a faculty startup grant from Iowa State University. Computational resources are provided by Meta and HPC@ISU. The resources from HPC@ISU include equipment funded by the U.S. National Science Foundation under MRI Grant Nos. 1726447 and 2018594.

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

## Appendix

Table 4: Pass@1 accuracy and model size of Llama 2-7B compressed with various low-rank algorithms on the mathematical reasoning task.

|       |                       |      |      |      |      |      |      |      |
|-------|-----------------------|------|------|------|------|------|------|------|
| SVD   | Model Size (billion)  | 6.74 | 5.02 | 3.18 | 1.73 | 1.11 | 0.56 |      |
|       | GSM8K Acc (%)         | 66.4 | 63.0 | 61.0 | 53.9 | 32.9 | 11.9 |      |
|       | MATH Acc (%)          | 20.6 | 18.3 | 17.4 | 13.7 | 5.3  | 2.8  |      |
| FWSVD | Model Size (billion)  | 6.74 | 4.79 | 2.95 | 1.54 | 0.96 | 0.47 |      |
|       | GSM8K Acc (%)         | 66.4 | 62.7 | 62.5 | 56.5 | 1.5  | 1.9  |      |
|       | MATH Acc (%)          | 20.6 | 19.2 | 17.6 | 14.2 | 1.8  | 1.5  |      |
| Basel | Model Size (billion)  | 6.74 | 5.14 | 3.28 | 1.83 | 1.21 | 0.67 | 0.43 |
|       | GSM8K Acc (%)         | 66.4 | 63.8 | 61.6 | 56.2 | 50.2 | 40.2 | 34.2 |
|       | MATH Acc (%)          | 20.6 | 19.7 | 17.6 | 15.5 | 13.4 | 10.0 | 7.8  |

Table 5: Pass@1 accuracy and model size of Llama 2-7B compressed with various pruning algorithms on the mathematical reasoning task.

|       |                       |      |      |      |      |      |      |      |
|-------|-----------------------|------|------|------|------|------|------|------|
| FLAP  | Model Size (billion)  | 6.74 | 5.39 | 4.72 | 4.04 | 3.37 | 2.70 | 1.35 |
|       | GSM8K Acc (%)         | 66.4 | 55.0 | 40.7 | 24.8 | 9.10 | 0    | 0    |
|       | MATH Acc (%)          | 20.6 | 14.5 | 8.4  | 4.2  | 1.5  | 0    | 0    |
| Wanda | Model Size (billion)  | 6.74 | 5.39 | 4.72 | 4.04 | 3.37 | 2.70 | 1.35 |
|       | GSM8K Acc (%)         | 66.4 | 61.0 | 44.6 | 19.0 | 0.5  | 0.3  | 0    |
|       | MATH Acc (%)          | 20.6 | 16.7 | 12.1 | 3.5  | 0.1  | 0.1  | 0    |

Table 6: Pass@1 accuracy and model size of Llama 2-13B compressed with various low-rank algorithms on the mathematical reasoning task.

| | Model Size (billion) | 13.02 | 9.70 | 6.10 | 3.27 | 2.07 | 1.01 | |
|---|---|---|---|---|---|---|---|---|
| SVD | GSM8K Acc (%) | 72.7 | 69.5 | 63.5 | 50.0 | 26.9 | 6.7 | |
| | MATH Acc (%) | 22.2 | 20.8 | 17.8 | 10.8 | 5.2 | 2.2 | |
| | Model Size (billion) | 13.02 | 9.24 | 5.67 | 2.93 | 1.79 | 0.83 | |
| FWSVD | GSM8K Acc (%) | 72.7 | 67.9 | 63.9 | 51.9 | 2.4 | 3.9 | |
| | MATH Acc (%) | 22.2 | 20.3 | 18.0 | 12.4 | 1.2 | 1.9 | |
| | Model Size (billion) | 13.02 | 9.75 | 6.13 | 3.32 | 2.14 | 1.12 | 0.68 |
| Basel | GSM8K Acc (%) | 72.7 | 67.9 | 64.4 | 55.0 | 50.3 | 41.8 | 37.9 |
| | MATH Acc (%) | 22.2 | 20.9 | 18.7 | 15.5 | 13.1 | 10.4 | 8.0 |

Table 7: Pass@1 accuracy and model size of Llama 2-13B compressed with various pruning algorithms on the mathematical reasoning task.

| | Model Size (billion) | 13.02 | 10.41 | 9.11 | 7.81 | 6.51 | 5.21 | 3.90 |
|---|---|---|---|---|---|---|---|---|
| FLAP | GSM8K Acc (%) | 72.7 | 60.7 | 51.2 | 4.6 | 0.2 | 0 | 0 |
| | MATH Acc (%) | 22.2 | 14.5 | 10.2 | 4.5 | 0.4 | 0 | 0 |
| | Model Size (billion) | 13.02 | 10.41 | 9.11 | 7.81 | 6.51 | 5.21 | 3.90 |
| Wanda | GSM8K Acc (%) | 72.7 | 66.0 | 50.3 | 0.1 | 0 | 0 | 0 |
| | MATH Acc (%) | 22.2 | 17.9 | 12.0 | 0.4 | 0.1 | 0 | 0 |

Table 8: Pass@1 accuracy and model size of Llama 2-7B compressed with various low-rank algorithms on the code generation task.

| | Model Size (billion) | 6.74 | 5.02 | 3.18 | 1.73 | 1.11 | 0.56 | |
|---|---|---|---|---|---|---|---|---|
| SVD | HumanEval Acc (%) | 23.8 | 20.7 | 20.1 | 9.1 | 4.9 | 3.7 | |
| | MBPP Acc (%) | 27.4 | 21.8 | 18.6 | 9.6 | 2.0 | 0.4 | |
| | Model Size (billion) | 6.74 | 4.84 | 3.01 | 1.58 | 0.99 | 0.49 | |
| FWSVD | HumanEval Acc (%) | 23.8 | 22.0 | 20.1 | 11.6 | 4.9 | 0 | |
| | MBPP Acc (%) | 27.4 | 24.4 | 17.4 | 10.4 | 0 | 0.6 | |
| | Model Size (billion) | 6.74 | 5.14 | 3.28 | 1.83 | 1.21 | 0.67 | 0.43 |
| Basel | HumanEval Acc (%) | 23.8 | 22.0 | 20.7 | 14.6 | 12.8 | 7.9 | 7.3 |
| | MBPP Acc (%) | 27.4 | 26.6 | 18.6 | 12.2 | 8.8 | 7.4 | 4.0 |

Table 9: Pass@1 accuracy and model size of Llama 2-7B compressed with various pruning algorithms on the code generation task.

| | Model Size (billion) | 6.74 | 5.40 | 4.72 | 4.04 | 3.37 | 2.70 | 1.35 |
|---|---|---|---|---|---|---|---|---|
| FLAP | HumanEval Acc (%) | 23.8 | 17.7 | 9.1 | 4.3 | 1.2 | 0.6 | 0 |
| | MBPP Acc (%) | 27.4 | 22.0 | 14.4 | 2.8 | 1.4 | 0.2 | 0 |
| | Model Size (billion) | 6.74 | 5.40 | 4.72 | 4.04 | 3.37 | 2.70 | 1.35 |
| Wanda | HumanEval Acc (%) | 23.8 | 18.9 | 15.9 | 3.7 | 0 | 0 | 0 |
| | MBPP Acc (%) | 27.4 | 24.4 | 19.4 | 7.2 | 0 | 0 | 0 |

Table 10: Pass@1 accuracy and model size of Llama 2-13B compressed with various low-rank algorithms on the code generation task.

| | | | | | | | | |
|---|---|---|---|---|---|---|---|---|
| **SVD** | Model Size (billion) | 13.02 | 9.70 | 6.10 | 3.27 | 2.07 | 1.01 | |
| | HumanEval Acc (%) | 27.4 | 18.9 | 18.3 | 3.0 | 3.7 | 0.6 | |
| | MBPP Acc (%) | 30.0 | 25.4 | 18.2 | 10.6 | 1.4 | 0.6 | |
| **FWSVD** | Model Size (billion) | 13.02 | 9.31 | 5.73 | 2.97 | 1.83 | 0.85 | |
| | HumanEval Acc (%) | 27.4 | 26.2 | 20.1 | 8.5 | 3.7 | 0 | |
| | MBPP Acc (%) | 30.0 | 27.2 | 21.6 | 12.2 | 0.8 | 0 | |
| **Basel** | Model Size (billion) | 13.02 | 9.75 | 6.13 | 3.32 | 2.14 | 1.12 | 0.68 |
| | HumanEval Acc (%) | 27.4 | 26.2 | 22.0 | 15.2 | 12.8 | 7.9 | 7.3 |
| | MBPP Acc (%) | 30.0 | 27.8 | 20.6 | 13.6 | 10.8 | 6.4 | 3.6 |

Table 11: Pass@1 accuracy and model size of Llama 2-13B compressed with various pruning algorithms on the code generation task.

| | | | | | | | | |
|---|---|---|---|---|---|---|---|---|
| **FLAP** | Model Size (billion) | 13.02 | 10.41 | 9.11 | 7.81 | 6.51 | 5.21 | 3.90 |
| | HumanEval Acc (%) | 27.4 | 17.1 | 15.9 | 5.5 | 2.4 | 0.6 | 0 |
| | MBPP Acc (%) | 30.0 | 20.8 | 16.0 | 5.6 | 2.8 | 0.6 | 0 |
| **Wanda** | Model Size (billion) | 13.02 | 10.41 | 9.11 | 7.81 | 6.51 | 5.21 | 3.90 |
| | HumanEval Acc (%) | 27.4 | 16.5 | 12.2 | 0 | 0 | 0 | 0 |
| | MBPP Acc (%) | 30.0 | 21.8 | 13.8 | 0 | 0 | 0 | 0 |

Table 12: Perplexity and model size of Llama-7B compressed with various low-rank algorithms on WikiText-2. The results of SVD, FWSVD, and SVD-LLM are cited from (Wang et al., 2025).

| | | | | | | | | |
|---|---|---|---|---|---|---|---|---|
| **SVD** | Model Size (billion) | 6.74 | 5.10 | 3.88 | 2.68 | 1.29 | | |
| | Perplexity | 5.68 | 20061 | 52489 | 105474 | 687291 | | |
| **FWSVD** | Model Size (billion) | 6.74 | 5.10 | 3.88 | 2.68 | 1.29 | | |
| | Perplexity | 5.68 | 1727 | 18156 | 32194 | 96872 | | |
| **SVD-LLM** | Model Size (billion) | 6.74 | 5.10 | 3.88 | 2.68 | 1.29 | | |
| | Perplexity | 5.68 | 7.73 | 9.27 | 15.00 | 31.79 | | |
| **Basel** | Model Size (billion) | 6.74 | 5.15 | 4.08 | 3.21 | 1.83 | 1.22 | 0.67 |
| | Perplexity | 5.68 | 6.11 | 6.68 | 6.99 | 7.93 | 9.21 | 10.45 |

Table 13: Pass@1 accuracy and model size of Llama 2-7B compressed and 8-bit-quantized by various algorithms for mathematical reasoning.

| 8-bit FLAP | Model Size (GB) | 6.74 | 5.39 | 4.72 | 4.04 | 3.37 | 2.70 | 1.35 |
|---|---|---|---|---|---|---|---|---|
| | GSM8K Acc (%) | 66.0 | 53.4 | 39.1 | 22.7 | 7.4 | 0 | 0 |
| | MATH Acc (%) | 20.3 | 13.5 | 7.2 | 4.4 | 1.0 | 0 | 0 |
| 8-bit Wanda | Model Size (GB) | 6.74 | 5.39 | 4.72 | 4.04 | 3.37 | 2.70 | 1.35 |
| | GSM8K Acc (%) | 66.0 | 59.7 | 44.4 | 15.8 | 0.3 | 0.2 | 0 |
| | MATH Acc (%) | 20.3 | 16.3 | 11.3 | 2.7 | 0.1 | 0 | 0 |
| 8-bit Basel | Model Size (GB) | 6.74 | 5.14 | 3.28 | 1.83 | 1.21 | 0.67 | 0.43 |
| | GSM8K Acc (%) | 66.0 | 62.1 | 59.6 | 54.1 | 49.7 | 41.1 | 35.3 |
| | MATH Acc (%) | 20.3 | 18.2 | 18.1 | 15.7 | 13.1 | 10.1 | 7.5 |

Table 14: Pass@1 accuracy and model size of Llama 2-13B compressed and 8-bit-quantized by various algorithms for mathematical reasoning.

| 8-bit FLAP | Model Size (GB) | 13.02 | 10.41 | 9.11 | 7.81 | 6.51 | 5.21 | 3.90 |
|---|---|---|---|---|---|---|---|---|
| | GSM8K Acc (%) | 72.7 | 59.5 | 53.9 | 6.1 | 0.2 | 0 | 0 |
| | MATH Acc (%) | 21.8 | 13.9 | 9.4 | 4.8 | 0.1 | 0 | 0 |
| 8-bit Wanda | Model Size (GB) | 13.02 | 10.41 | 9.11 | 7.81 | 6.51 | 5.21 | 3.90 |
| | GSM8K Acc (%) | 72.7 | 66.4 | 50.5 | 0.2 | 0 | 0 | 0 |
| | MATH Acc (%) | 21.8 | 17.0 | 10.7 | 0.5 | 0.1 | 0 | 0 |
| 8-bit Basel | Model Size (GB) | 13.02 | 9.75 | 6.13 | 3.32 | 2.14 | 1.12 | 0.68 |
| | GSM8K Acc (%) | 72.7 | 67.4 | 61.9 | 53.4 | 51.3 | 40.7 | 35.9 |
| | MATH Acc (%) | 21.8 | 19.3 | 17.5 | 14.2 | 12.5 | 9.6 | 7.7 |

Table 15: Pass@1 accuracy and model size of Llama 2-7B compressed and 8-bit-quantized by various algorithms for code generation.

| 8-bit FLAP | Model Size (GB) | 6.74 | 5.39 | 4.72 | 4.04 | 3.37 | 2.70 | 1.35 |
|---|---|---|---|---|---|---|---|---|
| | HumanEval Acc (%) | 23.8 | 16.5 | 9.1 | 3.7 | 1.2 | 0 | 0 |
| | MBPP Acc (%) | 26.2 | 20.8 | 12.4 | 3.4 | 1.2 | 0.2 | 0 |
| 8-bit Wanda | Model Size (GB) | 6.74 | 5.39 | 4.72 | 4.04 | 3.37 | 2.70 | 1.35 |
| | HumanEval Acc (%) | 23.8 | 16.5 | 13.4 | 3.7 | 0 | 0 | 0 |
| | MBPP Acc (%) | 26.2 | 22.2 | 18.2 | 4.6 | 0 | 0 | 0 |
| 8-bit Basel | Model Size (GB) | 6.74 | 5.14 | 3.28 | 1.21 | 0.67 | 0.43 | |
| | HumanEval Acc (%) | 23.8 | 20.1 | 19.5 | 11.6 | 9.1 | 7.3 | |
| | MBPP Acc (%) | 26.2 | 22.4 | 17.2 | 8.2 | 4.6 | 3.2 | |

Table 16: Pass@1 accuracy and model size of Llama 2-13B compressed and 8-bit-quantized by various algorithms for code generation.

| | | | | | | | | |
|---|---|---|---|---|---|---|---|---|
| 8-bit FLAP | Model Size (GB) | 13.02 | 10.41 | 9.11 | 7.81 | 6.51 | 5.21 | 3.90 |
| | HumanEval Acc (%) | 25.0 | 17.7 | 7.9 | 4.9 | 3.2 | 0.6 | 0 |
| | MBPP Acc (%) | 29.0 | 18.8 | 12.0 | 6.8 | 3.2 | 0.4 | 0 |
| 8-bit Wanda | Model Size (GB) | 13.02 | 10.41 | 9.11 | 7.81 | 6.51 | 5.21 | 3.90 |
| | HumanEval Acc (%) | 25.0 | 16.5 | 12.2 | 0 | 0 | 0 | 0 |
| | MBPP Acc (%) | 29.0 | 20.6 | 14.2 | 0 | 0 | 0 | 0 |
| 8-bit Basel | Model Size (GB) | 13.02 | 9.75 | 6.13 | 3.32 | 2.14 | 1.12 | 0.68 |
| | HumanEval Acc (%) | 25.0 | 25.0 | 15.9 | 12.2 | 11.0 | 9.1 | 6.1 |
| | MBPP Acc (%) | 29.0 | 26.4 | 18.4 | 10.4 | 9.2 | 5.4 | 3.4 |

Table 17: Pass@1 accuracy and model size of Llama 2-7B and -13B 4-bit-quantized by QLoRA for mathematical reasoning and code generation.

| Model | Task | Model size (GB) | Accuracy (%) |
|---|---|---|---|
| 4-bit QLoRA 7B | GSM8K | 3.45 | 54.1 |
| | MATH | 3.45 | 12.6 |
| | HumanEval | 3.45 | 12.8 |
| | MBPP | 3.45 | 16.2 |
| 4-bit QLoRA 13B | GSM8K | 6.63 | 58.8 |
| | MATH | 6.63 | 13.2 |
| | HumanEval | 6.63 | 13.4 |
| | MBPP | 6.63 | 18.6 |

Table 18: Pass@1 accuracy and model size of 4-bit-quantized Llama 2-7B and Llama 2-13B for mathematical reasoning and code generation.

| Model | Task | Model size (GB) | Accuracy (%) |
|---|---|---|---|
| 4-bit-quantized 7B | GSM8K | 3.37 | 39.2 |
| | MATH | 3.37 | 8.8 |
| | HumanEval | 3.37 | 6.7 |
| | MBPP | 3.37 | 10.8 |
| 4-bit-quantized 13B | GSM8K | 6.51 | 52.8 |
| | MATH | 6.51 | 12.5 |
| | HumanEval | 6.51 | 8.5 |
| | MBPP | 6.51 | 14.2 |

Table 19: Ablation study: Impact of varying the additional dimension in Basel on the compression of Llama 2-7B for the mathematical reasoning task.

| | Model Size (billion) | 6.74 | 5.14 | 3.28 | 1.83 | 1.21 | 0.67 | 0.43 |
|---|---|---|---|---|---|---|---|---|
| Addt'l Dim 32 | GSM8K Acc (%) | 66.4 | 63.8 | 61.6 | 56.2 | 50.2 | 40.2 | 34.2 |
| | MATH Acc (%) | 20.6 | 19.7 | 17.6 | 15.5 | 13.4 | 10.0 | 7.8 |
| | Model Size (billion) | 6.74 | 5.06 | 3.20 | 1.75 | 1.13 | 0.59 | |
| Addt'l Dim 0 | GSM8K Acc (%) | 66.4 | 64.4 | 61.6 | 55.4 | 48.3 | 34.9 | |
| | MATH Acc (%) | 20.6 | 19.3 | 17.4 | 15.1 | 12.2 | 7.1 | |

Table 20: Ablation study: Impact of varying the pruning times in Basel on the compression of Llama 2-7B for the mathematical reasoning task.

| | Model Size (billion) | 6.74 | 5.14 | 3.28 | 1.83 | 1.21 | 0.67 | 0.43 |
|---|---|---|---|---|---|---|---|---|
| 100 times | GSM8K Acc (%) | 66.4 | 63.8 | 61.6 | 56.2 | 50.2 | 40.2 | 34.2 |
| | MATH Acc (%) | 20.6 | 19.7 | 17.6 | 15.5 | 13.4 | 10.0 | 7.8 |
| | Model Size (billion) | 6.74 | 5.11 | 3.27 | 1.82 | 1.20 | 0.65 | |
| 2 times | GSM8K Acc (%) | 66.4 | 62.9 | 59.1 | 54.4 | 47.2 | 32.1 | |
| | MATH Acc (%) | 20.6 | 18.6 | 18.1 | 14.5 | 11.0 | 6.2 | |

Table 21: Ablation study: Pass@1 accuracy and model size of Llama 2-7B Basel (proposed) and Basel (free basis) on the mathematical reasoning task.

| | Model Size (billion) | 6.74 | 5.14 | 3.28 | 1.83 | 1.21 | 0.67 | 0.43 |
|---|---|---|---|---|---|---|---|---|
| Basel (Proposed) | GSM8K Acc (%) | 66.4 | 63.8 | 61.6 | 56.2 | 50.2 | 40.2 | 34.2 |
| | MATH Acc (%) | 20.6 | 19.7 | 17.6 | 15.5 | 13.4 | 10.0 | 7.8 |
| | Model Size (billion) | 6.74 | 5.14 | 4.14 | 2.51 | 1.21 | 0.67 | 0.43 |
| Basel (Free Basis) | GSM8K Acc (%) | 66.4 | 61.6 | 59.8 | 56.7 | 44.7 | 29.0 | 16.8 |
| | MATH Acc (%) | 20.6 | 17.9 | 17.1 | 14.9 | 10.3 | 6.3 | 2.6 |

Table 22: Ablation study: Impact of incorporating $L1$ regularization in Basel on the compression of Llama 2-7B for the mathematical reasoning task.

| | Model Size (billion) | 6.74 | 5.14 | 3.28 | 1.83 | 1.21 | 0.67 | 0.43 |
|---|---|---|---|---|---|---|---|---|
| Basel (Proposed) | GSM8K Acc (%) | 66.4 | 63.8 | 61.6 | 56.2 | 50.2 | 40.2 | 34.2 |
| | MATH Acc (%) | 20.6 | 19.7 | 17.6 | 15.5 | 13.4 | 10.0 | 7.8 |
| | Model Size (billion) | 6.74 | 5.14 | 3.28 | 1.83 | 1.21 | 0.67 | 0.43 |
| Basel ($L1$ norm, $\lambda = 0.02$) | GSM8K Acc (%) | 66.4 | 61.8 | 61.2 | 55.2 | 49.4 | 39.3 | 33.7 |
| | MATH Acc (%) | 20.6 | 18.8 | 17.8 | 15.3 | 12.5 | 8.9 | 8.1 |
| | Model Size (billion) | 6.74 | 5.13 | 3.28 | 1.83 | 1.21 | 0.67 | 0.43 |
| Basel ($L1$ norm, $\lambda = 0.1$) | GSM8K Acc (%) | 66.4 | 63.2 | 61.9 | 54.1 | 50.0 | 40.9 | 21.2 |
| | MATH Acc (%) | 20.6 | 19.5 | 17.5 | 15.0 | 12.4 | 8.8 | 3.8 |

