# OpenReview forum: "Streamlining Language Models via Semantic Basis Analysis"
_TMLR — Accepted by TMLR_

### Review · Reviewer_mkFe · 2025-09-01

**Summary Of Contributions:**

This paper proposes a method for reducing the size of pretrained language models when adapting them for downstream applications, by using a SVD decomposition of their internal weights. The method involves fixing the orthogonal SVD basis vectors of the original model, but relearning the singular values, and combining this with an auxiliary low-rank additive component. The model is then iteratively pruned and fine-tuned on the downstream task, dropping basis vectors from the original model that have small singular values after each fine-tuning stage.

The authors demonstrate that their method is more effective than directly doing SVD-based pruning on a fine-tuned model, even when SVD is done using a task-aware weighting (FWSVD), and enables models to retain more of their performance even at substantial compression ratios. They also show that it can be combined with quantization, and that in this case it outperforms some non-SVD based methods involving LoRAs or pruning.

**Additional Comments:**

I have a few other questions for the authors about their method:

1. The analysis in the first part of Section 3 studies the factorization of the $W^h_O W^h_V$ matrix of the attention layers (which I assume means the product of these). However, the description on page 5 says that Basel prunes "the $W\_Q$, $W\_K$, $W\_V$, and $W\_O$ matrices in the attention layers". Does this mean that $W\_O$ and $W\_V$ are pruned independently, with independent SVD decompositions, even though the method was motivated by the SVD of the product of these?
    - If so, it might be worth adding a brief footnote about this. I would also be curious if you have tried decomposing them jointly, and how well that performs if so.
2. I am curious if you have considered regularizing the re-learned $\tilde{s}_i$ weights using something like L1 regularization to encourage them to become sparse, as an alternative to (or in addition to) the iterative pruning process? This might guide the optimization process toward selecting lower-rank subsets of bases on its own.

**Audience:**

Yes

**Audience Explanation:**

The method appears to be quite effective and is straightforward to implement, is novel to the best of my knowledge, and focuses on a relevant and timely application area. I think some of TMLR's audience would be quite interested in learning about this method and the results presented in the paper.

**Broader Impact Concerns:**

No broader impact concerns.

**Claims And Evidence:**

Yes

**Claims Explanation:**

[Edit: Changed from "no" to "yes" after revision. Original review below]

There are two aspects of the paper as currently written that I think are not fully supported by evidence (although I believe both can be addressed with fairly minor revisions).

### 1. Claims about "bases encode highly specific semantic content"
The paper claims that "Our analysis reveals that the bases of these weight matrices encode distinct semantic components, some of which are redundant for specific target applications" and states that "bases encode highly specific semantic content". However, the evidence for this claim is quite weak: the authors present a table of five basis vectors, along with the ten tokens whose unembedding dot-product with these are the highest, and then categorize them based on what those ten tokens look like.

I think this claim is too strong for two reasons:
1. Knowing that a basis vector has high cosine similarity with some tokens related to a semantic concept (e.g. "iOS" and "Xcode") does **not** imply that this basis vector actually "encodes" this concept. Figuring out the semantic meaning of directions in a model is a complex research problem in its own right, and this kind of claim is very susceptible to confirmation bias. (Not exactly the same setting, but see [this paper](https://arxiv.org/abs/2506.05774) for context on how difficult it can be to justify this kind of claim in practice.)
2. It is not at all clear from the evidence provided that the basis vectors are actually "highly specific". For one, there are only five basis vectors presented, and it is not clear to me how they were chosen. (Were they cherry-picked for being the most specific-looking?) Also, I would guess that many random directions in activation space would have similar-looking top-ten tokens, because the tokens themselves are clustered in embedding space. Determining whether the basis vectors are _actually_ highly specific would require doing some sort of fair comparison between the basis directions and some other set of directions (e.g. random directions in the output space of the matrix), and possibly also evaluating the specificity based on something beyond the top-ten token list.

I don't think this is a major issue with the paper, because this part of the paper seems mostly to be used as motivation for the technique, which works well. But I think this claim should either be weakened substantially or supported with more evidence.

### 2. Comparisons to non-low-rank compression methods
The paper states that the method is more effective than "state-of-the-art techniques", and in section 4.1 it mentions comparing to SVD, FWSVD, QLoRA, and FLAP. However, the main experiments in the paper seem to only compare against SVD and FWSVD; the comparisons to QLoRA and FLAP are only done in combination with quantization.

It seems like quantization might affect the results of this comparison (especially given that FLAP was proposed in the absence of quantization), so it seems important to also compare to non-SVD-based methods in the unquantized setting. (I also think the paper would be improved by comparing to more than one pruning baseline, although this is not strictly required for the "supported by accurate, convincing and clear evidence" criterion.)

**Requested Changes:**

### Qualify claims about the bases "encoding" certain concepts and being "highly specific" (or add more supporting evidence)

The authors should weaken/qualify the claims in the paper about the bases found by SVD. In particular, I think saying that the bases "encode" semantic concepts, or that they are "highly specific", are not justified from the evidence provided. I would suggest using weaker and more correlation-based terminology, e.g. that the bases _are associated with_ semantic concepts, or that they _seem to correspond to_ particular semantic concepts. Similarly I think the authors should avoid saying that "the non-English character basis" is "irrelevant" to math problems; it is possible that this basis direction is being used by the model for something else as well, and that the top-ten tokens are not representative of the function of this direction on a math dataset.

(Alternatively, if the authors prefer to keep these claims, they should add substantially more empirical evidence to justify them.)

### Compare to a pruning baseline (perhaps FLAP) in the main unquantized experiments (figures 3,4,5,6), or weaken claims about outperforming SOTA compression
Currently, the bulk of the experiments do not compare against non-SVD-based techniques, which are only used in the quantization setting. To support the claims that the method outperforms SOTA techniques, and ensure that quantization is not interfering with the results, I think the method should be compared against some non-SVD-based techniques in the unquantized setting.

(Optional: The paper would also be stronger if it compared to more non-SVD baselines than just QLoRA and FLAP. For instance, [TrimLLM](https://aclanthology.org/2025.acl-long.33.pdf) is a recent technique that drops entire layers from the model, based on somewhat similar motivation to this paper.)

---

> ### Author Response · Authors · 2025-09-29
> **Response to Reviewer mkFe**
>
> We highly appreciate the reviewer’s detailed and constructive feedback. We have carefully addressed all the points in the revised version, as detailed below.
>
> &nbsp;
>
> **Regarding Requested Change 1**
>
> >Qualify claims about the bases "encoding" certain concepts and being "highly specific" (or add more supporting evidence).
> >
> >The authors should weaken/qualify the claims in the paper about the bases found by SVD. In particular, I think saying that the bases "encode" semantic concepts, or that they are "highly specific", are not justified from the evidence provided. I would suggest using weaker and more correlation-based terminology, e.g. that the bases are associated with semantic concepts, or that they seem to correspond to particular semantic concepts. Similarly I think the authors should avoid saying that "the non-English character basis" is "irrelevant" to math problems; it is possible that this basis direction is being used by the model for something else as well, and that the top-ten tokens are not representative of the function of this direction on a math dataset.
> >
> >(Alternatively, if the authors prefer to keep these claims, they should add substantially more empirical evidence to justify them.)
>
>
> In the revised version, we weakened our claims following the reviewer’s guidance. We removed overly strong  statements such as “bases encode semantic concepts,” “highly specific,” and “the non-English character basis is irrelevant to math problems.” Instead, we now adopt the softer terminology suggested by the reviewer, such as “is associated with” and “seem to correspond to.” These changes have been applied consistently throughout the paper, with particular adjustments in the abstract, introduction, related work, and Section 3.
>
> &nbsp;
>
> **Regarding Requested Change 2**
>
> >Compare to a pruning baseline (perhaps FLAP) in the main unquantized experiments (figures 3,4,5,6), or weaken claims about outperforming SOTA compression.
> >
> >Currently, the bulk of the experiments do not compare against non-SVD-based techniques, which are only used in the quantization setting. To support the claims that the method outperforms SOTA techniques, and ensure that quantization is not interfering with the results, I think the method should be compared against some non-SVD-based techniques in the unquantized setting.
> >
> >(Optional: The paper would also be stronger if it compared to more non-SVD baselines than just QLoRA and FLAP. For instance, TrimLLM is a recent technique that drops entire layers from the model, based on somewhat similar motivation to this paper.)
>
>
>
> In the revised version, we have added Wanda as an additional non-SVD baseline and now compare Basel against FLAP and Wanda in both unquantized and quantized settings. Our results show that Basel consistently outperforms these non-SVD baselines in both unquantized and quantized settings. The new comparisons in the unquantized setting are presented in Sections 4.2 and 4.3, with details in Figures 3–6 and Tables 5, 7, 9, and 11. The new comparisons in the quantized setting are presented in Sections 4.5 and 4.6, with details in Figures 8–11 and Tables 13-16.
>
> We have also removed all SOTA claims and now frame our results more rigorously: we state that Basel outperforms the evaluated baselines, rather than claiming superiority over all methods.
>
> In addition, we have cited the insightful related works mentioned by the reviewer, including [Hu, Rosing, and Zhang, ACL 25] and [Oikarinen, Yan, and Weng, ICML 25], in the revised version.
>
> &nbsp;
>
> **Regarding Additional Comment 1**
>
> We followed the reviewer’s suggestion and added a footnote (Footnote 2, Page 6) to clarify that matrices $W_O$ and $W_V$ are pruned independently. Due to computational resource constraints, exploring the joint pruning of $W_O$ and $W_V$ is left as future work, as noted in the same footnote.
>
> &nbsp;
>
> **Regarding Additional Comment 2**
>
> In the revised version, we have included additional experiments analyzing the effect of adding L1 regularization on top of Basel. These results are presented in the last paragraph of Section 4.8, as well as in Figure 16 and Table 22. We find that when the weight $\lambda$ of the L1 regularization is small ($\lambda = 0.02$), it has no noticeable impact on performance. However, with a larger weight ($\lambda = 0.1$), L1 regularization leads to decreased performance under deep compression.
>
> &nbsp;
>
> We wholeheartedly thank the reviewer once again for their valuable comments. We believe these revisions significantly improve the paper, and we are happy to provide further clarifications or adjustments if needed.

---

> > ### Comment · Reviewer_mkFe · 2025-10-05
> >
> > Thank you for the reply and for updating the paper. My concerns have been addressed, and I now believe the claims in the paper are sufficiently well supported by clear and convincing evidence.

---

> > > ### Author Response · Authors · 2025-10-06
> > >
> > > We sincerely thank the reviewer for the prompt and constructive feedback and for the suggested revisions that have greatly strengthened our paper.
> > >
> > > We are very encouraged to learn that the reviewer finds the claims in the revised paper now sufficiently well supported by clear and convincing evidence. If possible, we would be most grateful if the rating for the “claims and evidence” criterion could be updated to reflect this improvement.
> > >
> > > We once again appreciate the reviewer’s constructive feedback and kind consideration.

---

### Review · Reviewer_TuPk · 2025-09-12

**Summary Of Contributions:**

This paper introduces Basel, a new method for compressing LLMs for domain specific applications. The main idea is a truncated-SVD-like compression. The authors argue that the singular vectors often carry semantic meaning, and therefore in specific domains, certain singular vectors can be safely omitted. A repeated two-step procedure is therefore presented, where singular values are first re-learned, conditioned on the original orthogonal bases, and then a truncated SVD is performed. Basel is compared to a naive truncated SVD, QLoRA, and a 4-bit quantization, and shows generally superior performance for a given compression ratio.

## Strengths

- I'm generally positive about the core idea, which I find simple and intuitive (a gradient-based reweighted SVD, followed by truncation, iteratively executed). A further intuitive connection is built via the "semantic bases" experiment, which illustrates that orthonormal bases may indeed group to fairly specific concepts.

- Basel seems to work well in the experiments presented. For small compression ratios on the experiments presented, it is competitive with FWSVD, while maintaining performance for much more aggressive compression ratios.

## Weaknesses

- The claims regarding semantic bases are much overclaimed, in my opinion. There is anecdotal evidence presented, and I think claims like "these bases may sometimes carry semantic meaning" would be okay, but claims such as "our analysis reveals that the bases of these weight matrices encode distinct semantic components, some of which are redundant for specific target applications" are quite a bit stronger than the evidence presented.

- The baselines presented are fairly limited and weak. There are now many different approaches to fine tuning + compression/quantization/pruning/etc., and several more recent methods claim much better performance at high compression ratios.

- While illustrative, it is somewhat limiting to consider exclusively math and coding datasets, especially with small models that perform quite poorly on these benchmarks in the first place.

- I couldn't find any code available, which I think would greatly improve both the reproducibility of this study, and the usability of the method. I highly encourage the authors to include reproducible code.

**Additional Comments:**

Some minor comments:

- *Page 1, first paragraph (Large language models [...])*: This paragraph should have some citations regarding its claims.

- *Page 1, "as demonstrated by our interpretation results in Section 3 shows"*: this sentence is quite hard to read, and I would recommend revising.

- *Page 1, last paragraph, "our interpretation results"*: this phrase is once again difficult to read.

- *Page 3, first paragraph, "the bases hold significant physical meanings"*: as detailed above, I do not believe this claim to be substantiated. However, even if substantiated, I think "semantic" was meant here, rather than "physical."

- *Page 3, above Equation (2)*: instead of writing $u_i \perp u_j$ etc., I would recommend something to the effect "$\langle u_i, u_j \rangle = \langle v_i, v_j \rangle = 0$ for $i \neq j$", as the inner product notation is used directly after, and $\perp$ is never properly introduced.

- *Page 3, equations*: there are many potential matrix norms and inner products. The Frobenius norm and inner product are used -- this should be noted somehow.

- *Algorithm 1*: Wherever equation (3) is referenced, "equation equation 3" is written. This occurs on lines 3, 5, and 16.

- *Algorithm 1*: "Tune the learnable parameters" is a bit vague -- I understand that this means ${\tilde s}_i$, ${\tilde u}_j$ and ${\tilde v}_j$ for $i = 1, \dots, r$ and $j = 1, \dots, {\tilde r}$, but I recommend writing this in the algorithm block for clarity.

## References

[1] Wang, X., Zheng, Y., Wan, Z., & Zhang, M. (2024). SVD-LLM: Truncation-aware singular value decomposition for large language model compression. ICLR 2025.

[2] Yuan, Z., Shang, Y., Song, Y., Wu, Q., Yan, Y., & Sun, G. (2023). ASVD: Activation-aware singular value decomposition for compressing large language models. arXiv preprint arXiv:2312.05821.

**Audience:**

Yes

**Audience Explanation:**

Yes, pruning of LLMs is clearly of interest to the TMLR audience.

**Claims And Evidence:**

No

**Claims Explanation:**

As discussed above, there are a few concrete points where I think the paper lacks evidence:

- I do not think that the strong and general claims regarding the interpretability of the weight matrices are substantiated. The example presented is motivating, but anecdotal. I suspect that showing the semantic meaning more generally is quite difficult, and would suggest a weakening of claims, treating this as a motivating factor, rather than a central claim.

- Claims with respect to state-of-the-art seem lacking, as some reasonable baselines are not included. For example, SVD-LLM [1] and ASVD+ [2] are both recent, and demonstrate improved performance at high compression ratios.

**Requested Changes:**

- [Critical] As discussed above, the inclusion of several more recent and more competitive baselines, such as [1], [2].
- [Critical] As discussed above, either further evidence, or a weakening of claims, surrounding the interpretation of "semantic bases."
- [Critical] Discussion of hyperparameters used for the experiments. For example, what value of ${\tilde r}$ is used? Same for IterationsPerPruning, KeepingEpoch, PruningEpoch, etc.
- As discussed above, experiments in a wider range of fine-tuning settings, other than just math and code generation, potentially with larger models.
- Inclusion of reproducible code.

---

> ### Author Response · Authors · 2025-09-29
> **Response to Reviewer TuPk**
>
> We appreciate the reviewer’s thorough feedback. We have thoroughly addressed all of the requested changes in the revised manuscript and in our response below.
>
> &nbsp;
>
>
> **Regarding Requested Changes 1 and 4**
> > [Critical] As discussed above, the inclusion of several more recent and more competitive baselines, such as [1], [2].
>
> > As discussed above, experiments in a wider range of fine-tuning settings, other than just math and code generation, potentially with larger models.
>
> In this revised version, we added SVD-LLM [1] as a new baseline and compared it against our approach, along with other baselines, on the new task WikiText-2 (suggested by Reviewer iAwZ). Our results show that Basel outperforms SVD-LLM, achieving more significant model size reduction while maintaining better performance on WikiText-2. These new results are presented in Section 4.
>
> For ASVD+ [2], we were unable to find released code (the code of ASVD is available, but the code of ASVD+ is not). We also note that ASVD+ is currently available as a preprint, but has not yet appeared in a peer-reviewed venue. We very much appreciate the reviewer drawing our attention to this insightful work, and we have included a citation to it in the revised version.
>
> We also revised our claims to make them more rigorous. In the new version, we removed all SOTA claims (though our results do show that Basel outperforms SVD-LLM published at ICLR 2025), and we no longer state that our approach outperforms baselines across all domains. Instead, we now frame all claims in the form: “Our approach outperforms baselines X, Y, Z on compressing models A, B, C on tasks α, β, γ.” These baselines, models, and tasks are carefully matched to our experimental results. After this revision, all claims are fully supported by evidence. This aligns with the [TMLR acceptance criteria](https://jmlr.org/tmlr/acceptance-criteria.html), which recommend that any gap between claims and evidence should be addressed by either running more experiments or reducing claims.
>
> &nbsp;
>
> **Regarding Requested Change 2**
>
> > [Critical] As discussed above, either further evidence, or a weakening of claims, surrounding the interpretation of "semantic bases."
>
> In the revised version, we have weakened our claims regarding the interpretation of semantic bases. Following the reviewer’s suggestion, we removed statements such as “our analysis reveals that the bases of these weight matrices encode distinct semantic components, some of which are redundant for specific target applications.” Instead, we now use the softer wording suggested by the reviewer and Reviewer mkFe (e.g., “carry,” “is associated with,” “may correspond to”). These adjustments have been made consistently throughout the paper, especially in the abstract, introduction, related work, and Section 3.
>
> &nbsp;
>
> **Regarding Requested Change 3**
>
>  > [Critical] Discussion of hyperparameters used for the experiments. For example, what value of $\tilde{r}$ is used? Same for IterationsPerPruning, KeepingEpoch, PruningEpoch, etc.
>
> In the revised version, we provide the hyperparameters used in our experiments in Section 4.
>
> &nbsp;
>
> **Regarding Requested Change 5**
>
> > Inclusion of reproducible code.
>
> We are fully committed to releasing the code. Due to compliance requirements from the legal team of one of our affiliated institutions, the code can only be released upon acceptance of the paper. While this prevents us from sharing it at the submission stage, we give our firm assurance to the reviewers, the editor, and the research community that the code will be made publicly available once the paper is accepted. As confirmed in the [TMLR FAQ](https://jmlr.org/tmlr/faq.html), “Code and videos may be uploaded and associated with camera-ready papers.” Consistent with this policy, we will ensure that the code is released at the camera-ready stage.
>
> &nbsp;
>
> **Regarding Additional Comments**
>
> > Some minor comments:
>
> We appreciate the reviewer’s detailed comments. We have carefully addressed all of them in this revised version, with corresponding changes made to the introduction, related work, and Section 3.
>
> &nbsp;
>
> We wholeheartedly thank the reviewer once again for their valuable comments. We believe these revisions significantly improve the paper, and we are happy to provide further clarifications or adjustments if needed.

---

> > ### Comment · Reviewer_TuPk · 2025-10-14
> >
> > I thank the authors for their constructive revisions, which I now believe are sufficiently supported by evidence. I hope the authors will follow through on their promise for reproducible code, which will significantly improve impact and scientific value.

---

### Review · Reviewer_iAwZ · 2025-09-15

**Summary Of Contributions:**

This paper proposes Basel, a method to prune LLMs by reinterpreting weight matrices via SVD. Compared to the vanilla SVD pruning method, Basel relearns basis on target task. The approach is evaluated on LLaMA-2 (7B, 13B) across math (GSM8K, MATH) and code (HumanEval, MBPP), and the results show less accuracy drop off at high compression ratios compared to other baselines.

**Audience:**

Yes

**Audience Explanation:**

The method is novel enough to differentiate from its variants whilst being interpretable.

**Claims And Evidence:**

Yes

**Claims Explanation:**

- The paper compares with 2 SVD pruning variants - SVD and FWSVD, at high compression. Claims of accuracy gains and model size reduction are well supported by tables/figures. The qualitative analysis of bases adds credibility.
- Section 4.8 makes a compelling argument with evidence.

**Requested Changes:**

- Since Basel requires an additional retraining phase (re-learning singular values) absent in SVD and FWSVD, can you quantify this computational overhead? I would assume it is much less than the full dense model but it is still nice to know.

- Broaden the experiment if possible. The current evaluation covers math and code, which are both specialized, knowledge-intensive tasks, but it would be nice to include a demonstration on a more conventional NLP dataset such as wikitext2. Even a brief result in the appendix would help illustrate that the method isn’t limited to the two domains chosen. If new experiments are not feasible, consider adding a sentence in the experiments section or conclusion noting that “Basel can in principle be applied to other domains; we chose math and code as challenging test cases, but expect similar benefits in NLP tasks, etc.” I consider this a nice-to-have because the paper is already strong; it’s not a deal-breaker, but would add value if doable.

---

> ### Author Response · Authors · 2025-09-29
> **Response to Reviewer iAwZ**
>
> We sincerely thank the reviewer for the insightful and constructive feedback. We have carefully revised the paper to incorporate the requested analysis.
>
> &nbsp;
>
> **Regarding Requested Change 1**
>
> > Since Basel requires an additional retraining phase (re-learning singular values) absent in SVD and FWSVD, can you quantify this computational overhead? I would assume it is much less than the full dense model but it is still nice to know.
>
> Indeed, the additional retraining in Basel incurs only a fraction of the cost compared to full dense model finetuning. Specifically, Basel requires approximately 30% of the GPU memory and 46% of the GPU hours relative to full finetuning. To make this point clear, we have added a detailed analysis of Basel’s computational overhead at the end of Section 3 in the revised paper.
>
> &nbsp;
>
> **Regarding Requested Change 2**
>
> > Broaden the experiment if possible. The current evaluation covers math and code, which are both specialized, knowledge-intensive tasks, but it would be nice to include a demonstration on a more conventional NLP dataset such as wikitext2. Even a brief result in the appendix would help illustrate that the method isn’t limited to the two domains chosen. If new experiments are not feasible, consider adding a sentence in the experiments section or conclusion noting that “Basel can in principle be applied to other domains; we chose math and code as challenging test cases, but expect similar benefits in NLP tasks, etc.” I consider this a nice-to-have because the paper is already strong; it’s not a deal-breaker, but would add value if doable.
>
> We thank the reviewer for this thoughtful suggestion. Following the recommendation, we have broadened our evaluation to include WikiText-2, a widely used benchmark for language modeling. On this dataset, Basel achieves up to 4× additional model size reduction compared to baseline methods, while also delivering superior performance. We have added a new section (Section 4.4) describing these results, with details provided in Figure 7 of the main text and Table 12 in the appendix. We believe this extension demonstrates that Basel is broadly applicable beyond math and code, further strengthening the paper.
>
> &nbsp;
>
> We wholeheartedly thank the reviewer once again for their valuable comments. We believe these revisions significantly improve the paper, and we are happy to provide further clarifications or adjustments if needed.

---

### Decision · Action_Editor_YiiH · 2025-11-06

**Recommendation:** Accept as is

**Additional Comments:**

There is unanimous support from the reviewers following the author rebuttal. Therefore, I recommend acceptance. Please ensure that the the link to the open-source code is included in the camera-ready version, as promised during rebuttal.

**Audience:**

Yes

**Audience Explanation:**

Yes. The paper presents a practical and effective method for reducing the size and cost of large language models while maintaining performance, which is a topic of broad relevance in the TMLR community.

**Claims And Evidence:**

Yes

**Claims Explanation:**

Yes. The proposed method is evaluated on two Llama models across math reasoning, code generation, and language modeling tasks, and compared against a wide range of baselines. The authors also provided an analysis of computational overhead, which further supports the efficiency claims.